# Precision gestational diabetes treatment: a systematic review and meta-analyses

Jamie L. Benham[1,198], Véronique Gingras[2,3,198], Niamh-Maire McLennan[4,5,198], Jasper Most [6,198], Jennifer M. Yamamoto[7,198], Catherine E. Aiken[8,9,198], Susan E. Ozanne [10,199], Rebecca M. Reynolds [4,5,199 ✉] & ADA/EASD PMDI*

**Abstract**

**Background** Gestational Diabetes Mellitus (GDM) affects approximately 1 in 7 pregnancies globally. It is associated with short- and long-term risks for both mother and baby. Therefore, optimizing treatment to effectively treat the condition has wide-ranging beneficial effects. However, despite the known heterogeneity in GDM, treatment guidelines and approaches are generally standardized. We hypothesized that a precision medicine approach could be a tool for risk-stratification of women to streamline successful GDM management. With the relatively short timeframe available to treat GDM, commencing effective therapy earlier, with more rapid normalization of hyperglycaemia, could have benefits for both mother and fetus. **Methods** We conducted two systematic reviews, to identify precision markers that may predict effective lifestyle and pharmacological interventions. **Results** There was a paucity of studies examining precision lifestyle-based interventions for GDM highlighting the pressing need for further research in this area. We found a number of precision markers identified from routine clinical measures that may enable earlier identification of those requiring escalation of pharmacological therapy (to metformin, sulphonylureas or insulin). This included previous history of GDM, Body Mass Index and blood glucose concentrations at diagnosis. **Conclusions** Clinical measurements at diagnosis could potentially be used as precision markers in the treatment of GDM. Whether there are other sensitive markers that could be identified using more complex individual-level data, such as omics, and if these can feasibly be implemented in clinical practice remains unknown. These will be important to consider in future studies.

**Plain language summary**

Gestational diabetes (GDM) is high blood sugar first detected during pregnancy. Normalizing blood sugar levels quickly is important to avoid pregnancy complications. Many women achieve this with lifestyle changes, such as to diet, but some need to inject insulin or take tablets. We did two thorough reviews of existing research to see if we could predict which women need medication. Firstly we looked for ways to identify the characteristics of women who benefit most from changing their lifestyles to treat GDM, but found very limited research on this topic. We secondly searched for characteristics that help identify women who need medication to treat GDM. We found some useful characteristics that are obtained during routine pregnancy care. Further studies are needed to test if additional information could provide even better information about how we could make GDM treatment more tailored for individuals during pregnancy.

[1] Department of Medicine and Community Health Sciences, Cumming School of Medicine, University of Calgary, Calgary, AB, Canada. [2] Department of Nutrition, Université de Montréal, Montreal, QC, Canada. [3] Research Center, Sainte-Justine University Hospital Center, Montreal, QC, Canada. [4] MRC Centre for Reproductive Health, Queens's Medical Research Institute, University of Edinburgh, Edinburgh, UK. [5] Centre for Cardiovascular Science, Queens's Medical Research Institute, University of Edinburgh, Edinburgh, UK. [6] Department of Orthopedics, Zuyderland Medical Center, Sittard-Geleen, The Netherlands. [7] Internal Medicine, University of Manitoba, Winnipeg, MB, Canada. [8] Department of Obstetrics and Gynaecology, the Rosie Hospital, Cambridge, UK. [9] NIHR Cambridge Biomedical Research Centre, University of Cambridge, Cambridge, UK. [10] University of Cambridge Metabolic Research Laboratories and MRC Metabolic Diseases Unit, Wellcome-MRC Institute of Metabolic Science, Cambridge, UK. [198] These authors contributed equally: Jamie L. Benham, Véronique Gingras, Niamh-Maire McLennan, Jasper Most, Jennifer M. Yamamoto, Catherine E. Aiken. [199] These authors jointly supervised this work: Susan E. Ozanne, Rebecca M. Reynolds. *A list of authors and their affiliations appears at the end of the paper. ✉email: r.reynolds@ed.ac.uk

Gestational diabetes (GDM) is the most common pregnancy complication, occurring in 3–25% of pregnancies globally[1]. GDM is associated with short- and long-term risks to both mothers and babies, including adverse perinatal outcomes, future obesity, type 2 diabetes and cardiovascular disease[1–3]. The landmark Australian Carbohydrate Intolerance Study in Pregnant Women (ACHOIS) demonstrated that effective treatment of GDM reduces serious perinatal morbidity[4].

Current treatment guidelines for management of GDM assume homogeneous treatment requirements and responses, despite the known heterogeneity of GDM aetiology[5–8]. Standard care includes diet and lifestyle advice at a multi-disciplinary clinic, home blood glucose monitoring at least four times per day, clinic reviews every 2 to 4 weeks, and then progression to pharmacological treatment with metformin, glyburide and/or insulin if glucose targets are not met. Around a third of women cannot maintain euglycaemia with lifestyle measures alone and require treatment escalation to a pharmacological agent[3]. Yet current treatment pathways often take 4–8 weeks to achieve glucose targets. This delay resulting in continued exposure to hyperglycaemia poses a risk of accelerated foetal growth[9,10]. Previous research has suggested that maternal characteristics including body mass index (BMI) $\geq 30\,kg/m^2$, family history of type 2 diabetes, prior history of GDM and higher glycated haemoglobin (HbA1c) increase the likelihood of need for insulin treatment in GDM[11], indicating the potential for risk-stratification of women to streamline successful GDM management. There is emerging evidence that precision biomarkers predict treatment response in type 2 diabetes, which has similar heterogeneity to GDM[12,13] and thus gives rationale to investigate whether a similar precision approach could be successful in optimising outcomes in GDM.

To address this knowledge gap, we conducted two systematic reviews of the available evidence for precision markers of GDM treatment. We aimed to determine which patient-level characteristics are precision markers for predicting (i) responses to personalised diet and lifestyle interventions delivered in addition to standard of care (ii) requirement for escalation of treatment in women treated with diet and lifestyle alone, and in women receiving pharmacological agents for the treatment of GDM. For both reviews we considered whether the precision markers predicted achieving glucose targets, as well as maternal and neonatal outcomes. The Precision Medicine in Diabetes Initiative (PMDI) was established in 2018 by the American Diabetes Association (ADA) in partnership with the European Association for the Study of Diabetes (EASD). The ADA/EASD PMDI includes global thought leaders in precision diabetes medicine who are working to address the burgeoning need for better diabetes prevention and care through precision medicine[14]. This systematic review is written on behalf of the ADA/EASD PMDI as part of a comprehensive evidence evaluation in support of the 2nd International Consensus Report on Precision Diabetes Medicine[15].

We find a paucity of studies examining precision lifestyle-based interventions for GDM highlighting the pressing need for further research in this area. We find a number of precision markers identified from routine clinical measures that may enable earlier identification of those requiring escalation of pharmacological therapy (to metformin, sulphonylureas or insulin). These findings suggest that clinical measurements at diagnosis could potentially be used as precision markers in the treatment of GDM. Whether there are other sensitive markers that could be identified using more complex individual-level data, such as omics, and if these can feasibly be implemented in clinical practice remains unknown and will be important to consider in future studies.

## Methods

The systematic reviews and meta-analyses were performed as outlined a priori in the registered protocols (PROSPERO registration IDs CRD42022299288 and CRD42022299402). The Preferred Reporting Items for Systematic reviews and Meta-Analyses (PRISMA) guidelines[16] were followed. Ethical approval was not required as these were secondary studies using published data.

**Literature searches, search strategies and eligibility criteria.** Search strategies for both reviews were developed based on relevant keywords in partnership with scientific librarians (see Supplementary Note 1 for full search strategies). We searched two databases (MEDLINE and EMBASE) for studies published from inception until January 1st, 2022. We also scanned the references of included manuscripts for inclusion as well as relevant reviews and meta-analyses published within the past two years for additional citations.

For both systematic reviews we included studies (randomised or non-randomised trials and observational studies) published in English and including women ≥16 years old with diagnosed GDM, as defined by the study authors. For the first systematic review (precision diet and lifestyle interventions), we included studies with any behavioural intervention using any approach (e.g., specific exercise, dietary interventions, motivational interviewing) that examined precision markers that could tailor a lifestyle intervention in a more precise way compared to a control group receiving standard care only. For the second systematic review (precision markers for escalation of pharmacological interventions to achieve glucose targets), we included studies investigating women with GDM that required escalation of pharmacological therapy (e.g., insulin, metformin, sulphonylurea) compared to women with GDM that achieved glucose targets with diet and lifestyle measures only, or women with GDM treated with oral agents that required progression to insulin to achieve glucose targets. For both reviews, we included any relevant reported outcomes; maternal (e.g., treatment adherence, hypertensive disorders of pregnancy, gestational weight gain, mode of birth), neonatal (e.g., birthweight, macrosomia, shoulder dystocia, preterm birth, neonatal hypoglycaemia, neonatal death), cost efficiency or acceptability. We excluded studies with a total sample size <50 participants to ensure sufficient data to interpret the effect of precision markers. We also excluded studies published before or during 2004, in order to consider studies with standard care similar to ACHOIS[4].

**Study selection and data extraction.** The results of our two searches were imported separately into Covidence software (Veritas Health Innovation, Australia, available at www.covidence.org) and duplicates were removed. Two reviewers independently reviewed identified studies. First, they screened titles and abstracts of all references identified from the initial search. In a second step, the full-text articles of potentially relevant publications were scrutinised in detail and inclusion criteria were applied to select eligible articles. Reason for exclusion at the full-text review stage was documented. Disagreement between reviewers was resolved through consensus by discussion with the group of authors.

Two reviewers independently extracted relevant information from each eligible study, using a pre-specified standardised extraction form. Any disagreement between reviewers was resolved as outlined above.

Data extracted included first author name, year of publication, country, study design, type and details of the intervention when applicable, number of cases/controls or cohort groups, total

number of participants and diagnostic criteria used for GDM. Extracted data elements also included outcomes measures, size of the association (Odds Ratio (OR), Relative Risk (RR) or Hazard Ratio (HR)) with corresponding 95% Confidence Interval (CI) and factors adjusted for, confounding factors taken into consideration and methods used to control covariates. We prioritised adjusted values where both raw and adjusted data were available. Details of precision markers (mean (standard deviation) for continuous variables or N (%) for categorical variables) including BMI (pre-pregnancy or during pregnancy), ethnicity, age, smoking status, comorbidities, parity, glycaemic variables (e.g., oral glucose tolerance test (OGTT) diagnostic values, HbA1c), timing of GDM diagnosis, history of diabetes or of GDM, and season were also extracted.

**Quality assessment (risk of bias and GRADE assessments).** We first assessed the quality and risk of bias of each individual study using the Joanna Briggs Institute (JBI) critical appraisal tools[17]. A Grading of Recommendations, Assessment, Development, and Evaluations (GRADE) approach was then used to review the total evidence for each precision marker, and the quality of the included studies to assign a GRADE certainty to this body of evidence (high, moderate, low and/or very low)[18]. Quality assessment was performed in duplicate and conflicts were resolved through consensus.

**Statistical analysis.** Where possible, meta-analyses were conducted using random effects models for each precision marker available. The pooled effect size (mean difference for continuous outcomes and ORs for categorical outcomes) with the corresponding 95% CI was computed. The heterogeneity of the studies was quantified using $I^2$ statistics, where $I^2 > 50\%$ represents moderate and $I^2 > 75\%$ represents substantial heterogeneity across studies. Publication bias was assessed with visual assessment of funnel plots. Statistical analyses were performed using Review Manager software [RevMan, Version 5.4.1, The Cochrane Collaboration, Copenhagen, Denmark].

As part of the diabetes scientific community, we are sensitive in using inclusive language, especially in relation to gender. However, the vast majority of original studies that the GDM precision medicine working groups reviewed used women as their terminology to describe their population, as GDM per definition occurs in pregnancy which can only occur in individuals that are female at birth. To be consistent with the original studies defined populations, we use the word 'women' in our summary of the evidence, current gaps and future perspectives, but fully acknowledge that not all individuals who experienced a pregnancy may self-identify as women at all times over their life course.

**Reporting summary.** Further information on research design is available in the Nature Portfolio Reporting Summary linked to this article.

## Results

**Study selection and study characteristics.** PRISMA flow charts (Figs. 1 and 2) summarise both searches and study selection processes.

For the first systematic review (precision approaches to diet and lifestyle interventions), we identified 2 eligible studies ($n = 2354$ participants), which were randomised trials from USA and Singapore (Supplementary Data 1)[19,20].

For the second systematic review (precision markers for escalation of pharmacological interventions to achieve target glucose levels), we identified 48 eligible studies ($n = 25,724$ participants) (Supplementary Data 2)[21–68]. There were 34 studies

($n = 23,831$ participants) investigating precision markers for escalation to pharmacological agent(s) in addition to standard care with diet and lifestyle advice. Of these, 29 studies ($n = 20,486$) reported escalation to insulin as the only option[21–49] and 5 ($n = 3345$) reported escalation to any medication (metformin, glyburide and/or insulin)[50–54]. There were 12 studies ($n = 1836$ participants) investigating precision markers for escalation to insulin when treatment with oral agents was not adequate to achieve target glucose levels. Initial treatment was with glyburide in 6 of these studies ($n = 527$)[55–60] and metformin in the other 6 studies ($n = 1142$)[61–66]. A further 2 eligible studies reported maternal genetic predictors of need for supplementary insulin after glyburide ($n = 117$ participants)[67] and maternal lipidome responses to metformin and insulin ($n = 217$ participants)[68].

The majority of included studies were observational in design. Most studies reported outcomes of singleton pregnancies. The studies were from a range of geographical locations: Europe (Belgium, Finland, France, Italy, Netherlands, Poland, Portugal, Spain, Sweden), Switzerland, Middle East (Israel, Qatar, United Arab Emirates), Australasia (Australia, New Zealand), North America/Latin America (Canada, USA and Brazil) and Asia (China, Malaysia, Japan). There were a range of approaches to GDM screening, choice of diagnostic test and diagnostic glucose thresholds.

**Quality assessment.** Study quality assessment is presented as an overall risk of bias for the studies included in the meta-analyses in Fig. 3 and as a heat map for quality assessment for each included study in Fig. 4. Most of the studies were rated as low risk of bias, as they adequately described how a diagnosis of GDM was assigned, defining inclusion and exclusion criteria, and reported the protocol for initiation of pharmacological therapy. Not all studies reported whether women received diet and lifestyle advice as standard care. Few studies reported whether the precision marker was measured in a valid and reliable way. Using the GRADE approach, the majority of precision markers were classified as having a low certainty of evidence with some classified as very low certainty (Tables 1 and 2). No publication bias (as ascertained by funnel plot analyses) was detected.

**Precision diet and lifestyle interventions in GDM.** Two studies examining different precision approaches to behavioural interventions were included in the first systematic review, so we present a narrative synthesis of the findings. Neither study examined whether a precision approach to specific lifestyle interventions facilitated achievement of glucose targets during pregnancy or improved outcomes that reflect glycaemic control during pregnancy such as macrosomia, large for gestational age, or neonatal hypoglycaemia.

In one study of women with GDM[19], the intervention was distribution of a tailored letter based on electronic health record data detailing gestational weight gain (GWG) recommendations (as defined by the Institute of Medicine). Receipt of this tailored letter increased the likelihood of meeting the end-of-pregnancy weight goal among women with normal pre-pregnancy BMI, but not among women with overweight or obese pre-pregnancy BMI. This study identified normal pre-pregnancy BMI as a precision marker for intervention success.

The second study[20] used a Web/Smart phone lifestyle coaching programme in women with GDM. Pre-intervention excessive GWG was evaluated as a potential precision marker for the response to the Web/Smart phone lifestyle coaching programme in preventing excess GWG. There was no difference between study arms with respect to either excess GWG or absolute GWG

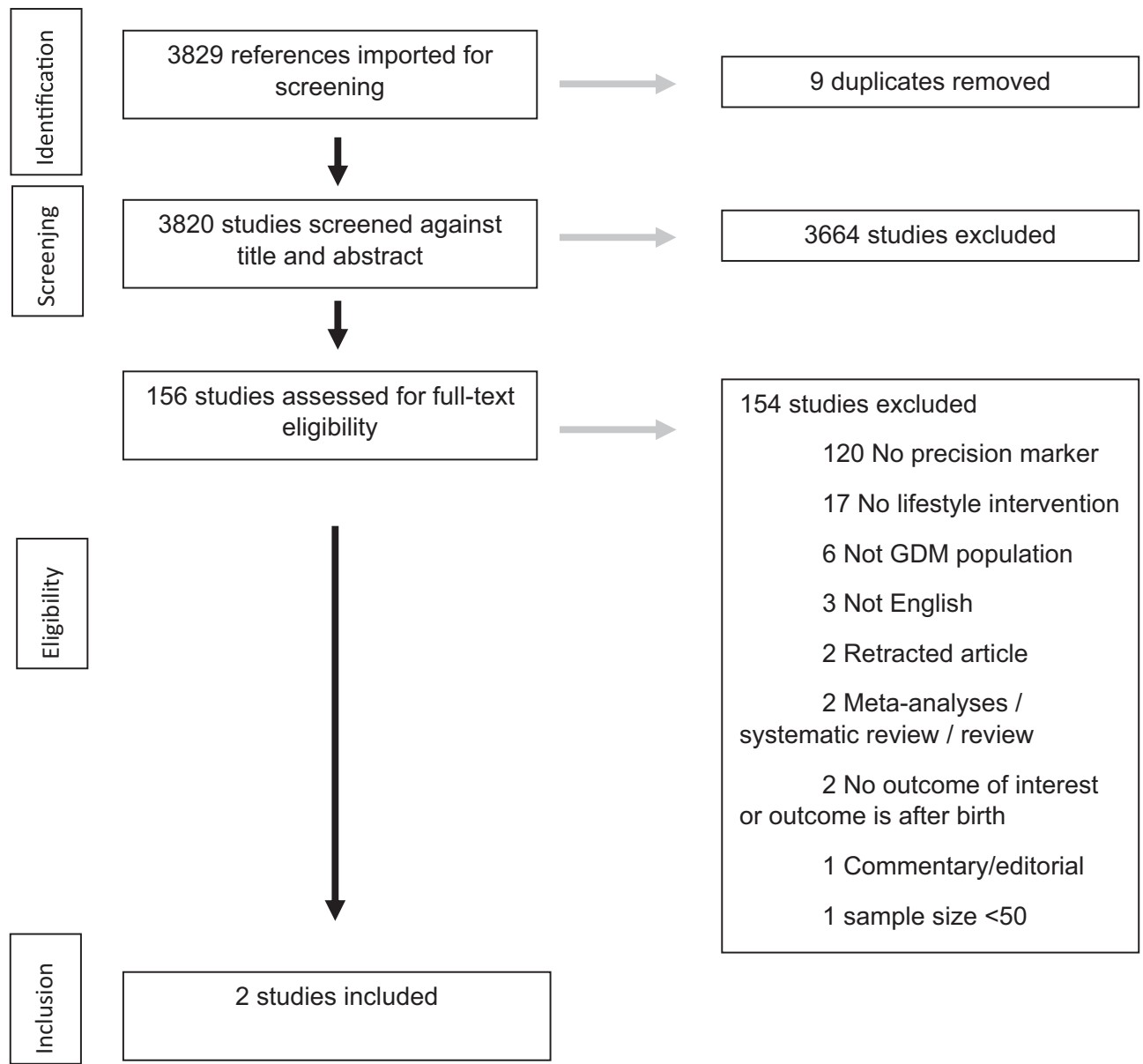

**Fig. 1 Preferred Reporting Items for Systematic Reviews and Meta-Analyses (PRISMA) flow diagrams for precision approaches to enhance behavioural (diet and lifestyle) interventions.** The PRISMA flow diagram details the search and selection process applied in the review.

by the end of pregnancy indicating that early GWG is not a useful precision marker with respect to this intervention.

**Precision markers for escalation of pharmacological interventions to achieve glucose targets in GDM.** Of the 34 studies of precision markers for escalation to pharmacological therapy to achieve glucose targets in addition to standard care with diet and lifestyle advice, 23 studies ($n = 19,112$ participants) were included in the meta-analysis[21–23,25,26,31–36,38,40,41,43–46,48,50–53] and 11 studies ($n = 7158$ participants) in the narrative synthesis[24,27–30,37,39,42,47,49,54].

Table 1 and Supplementary Figs. 1–13 show that precision markers for GDM to be adequately managed with lifestyle measures were lower maternal age, nulliparity, lower BMI, no previous history of GDM, lower HbA1c, lower glucose values at the diagnostic OGTT (fasting, 1 h, 2 and/or 3 h glucose), no family history of diabetes, later gestation of diagnosis of GDM and no

macrosomia in previous pregnancies. There was a similar pattern for not smoking but this did not reach statistical significance.

Twelve studies ($n = 1836$ participants) of precision markers for escalation to insulin to achieve glucose targets in addition to oral agents were included in the meta-analysis[55–66].

Table 2 and Supplementary Figs. 14–25 show that precision markers for achieving glucose targets with oral agents only were nulliparity, lower BMI, no previous history of GDM, lower HbA1c, lower glucose values at the diagnostic OGTT (fasting, 1 h, and/or 2 h glucose), later gestation of diagnosis of GDM and later gestation at initiation of the oral agent. In sensitivity analyses, there were no differences in the precision markers predicting response to metformin versus glyburide (Supplementary Data 3).

Similar precision markers for escalation to pharmacotherapy to achieve glucose targets were observed in the 11 studies ($n = 7158$ participants) that were not included in the meta-analysis[24,27–30,37,39,42,47,49,54] (Supplementary Data 4). Additional precision markers including foetal sex[28], ethnicity[30,47] and season

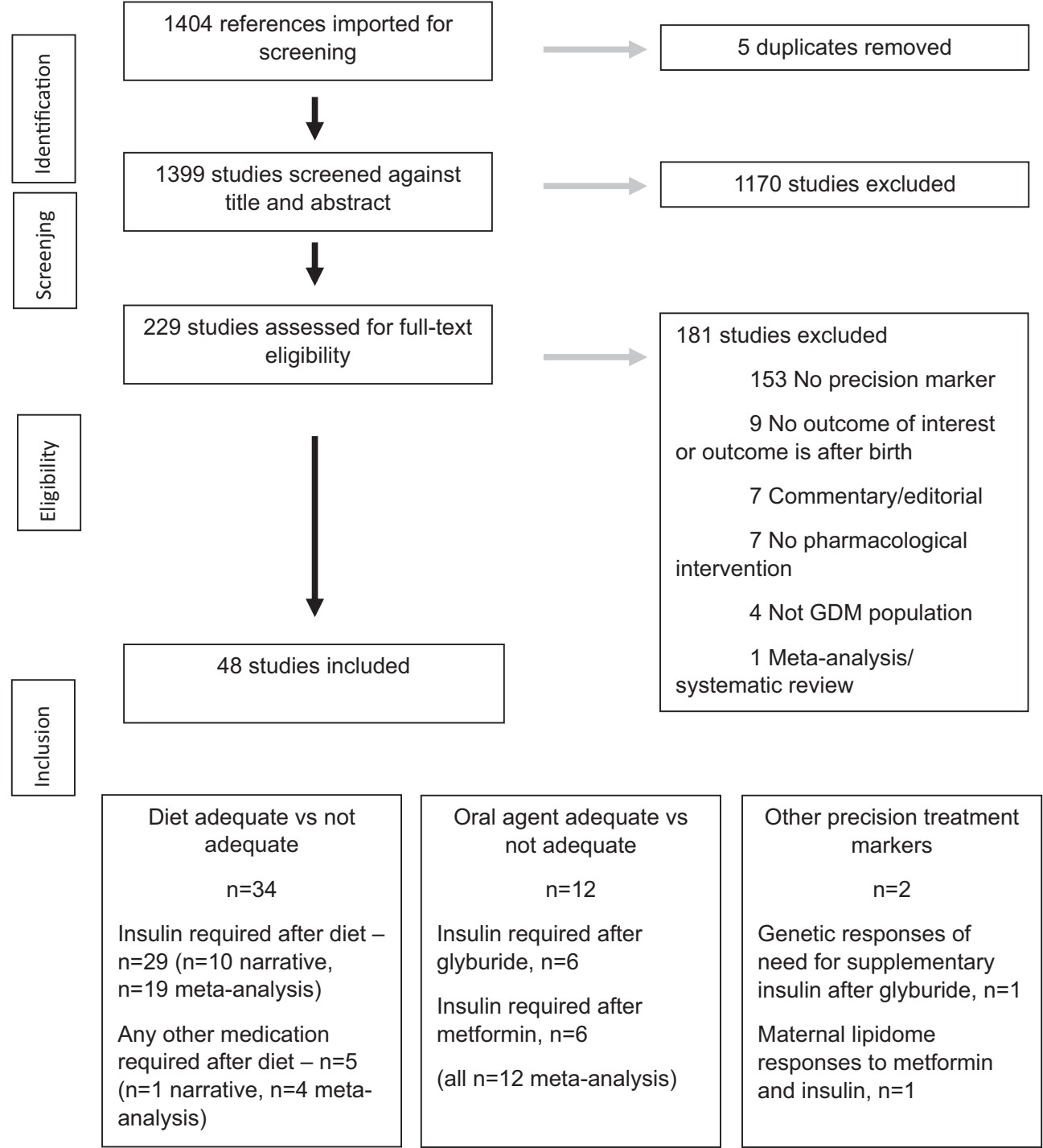

**Fig. 2 Preferred Reporting Items for Systematic Reviews and Meta-Analyses (PRISMA) flow diagrams for precision markers for escalation of pharmacological interventions.** The PRISMA flow diagram details the search and selection process applied in the review.

of birth[37] were evaluated in some studies but there was insufficient data to draw conclusions.

There was a paucity of data in examining other precision markers with only weak evidence that the maternal lipidome[68] or genetics[67] hold potential as precision markers for escalation of pharmacological treatment (Supplementary Data 4).

**Discussion**
As the factors contributing to the development of GDM and its aetiology are heterogeneous[5–8], it is plausible that the most effective treatment strategies may also be variable among women

with GDM. A precision medicine approach resulting in more rapid normalisation of hyperglycaemia could have substantial benefits for both mother and foetus. By synthesising the evidence from two systematic reviews, we sought to identify key precision markers that may predict effective lifestyle and pharmacological interventions. There were a paucity of studies examining precision approaches to better target lifestyle-based interventions for GDM treatment highlighting the pressing need for further research in this area. However, we found a number of precision markers to enable earlier identification of those requiring escalation of pharmacological therapy. These included characteristics

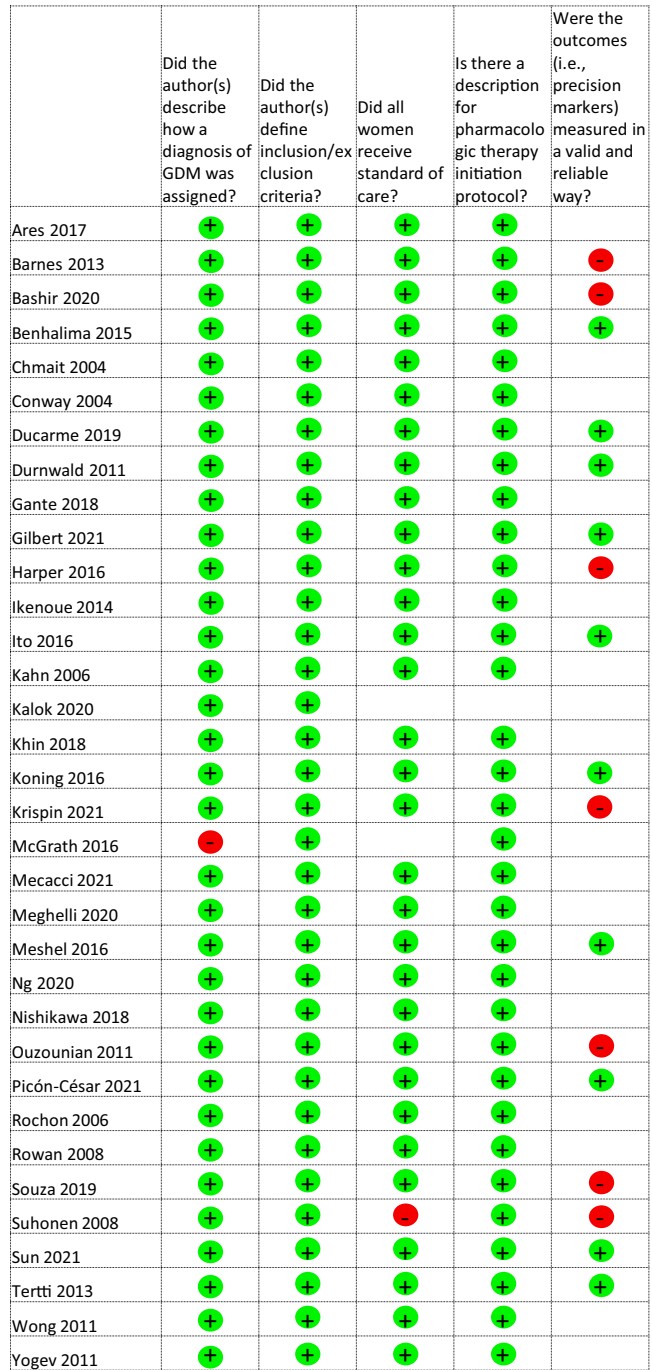

| | Did the author(s) describe how a diagnosis of GDM was assigned? | Did the author(s) define inclusion/exclusion criteria? | Did all women receive standard of care? | Is there a description for pharmacologic therapy initiation protocol? | Were the outcomes (i.e., precision markers) measured in a valid and reliable way? |
|---|---|---|---|---|---|
| Ares 2017 | + | + | + | + | |
| Barnes 2013 | + | + | + | + | − |
| Bashir 2020 | + | + | + | + | − |
| Benhalima 2015 | + | + | + | + | + |
| Chmait 2004 | + | + | + | + | |
| Conway 2004 | + | + | + | + | |
| Ducarme 2019 | + | + | + | + | + |
| Durnwald 2011 | + | + | + | + | + |
| Gante 2018 | + | + | + | + | |
| Gilbert 2021 | + | + | + | + | + |
| Harper 2016 | + | + | + | + | − |
| Ikenoue 2014 | + | + | + | + | |
| Ito 2016 | + | + | + | + | + |
| Kahn 2006 | + | + | + | + | |
| Kalok 2020 | + | + | | | |
| Khin 2018 | + | | + | + | |
| Koning 2016 | + | + | + | + | + |
| Krispin 2021 | + | + | + | + | − |
| McGrath 2016 | − | + | | + | |
| Mecacci 2021 | + | + | + | + | |
| Meghelli 2020 | + | + | + | + | |
| Meshel 2016 | + | + | + | + | + |
| Ng 2020 | + | + | + | + | |
| Nishikawa 2018 | + | + | + | + | |
| Ouzounian 2011 | + | + | + | + | − |
| Picón-César 2021 | + | + | + | + | + |
| Rochon 2006 | + | + | + | + | |
| Rowan 2008 | + | + | + | + | |
| Souza 2019 | + | + | + | + | − |
| Suhonen 2008 | + | + | − | + | − |
| Sun 2021 | + | + | + | + | + |
| Tertti 2013 | + | + | + | + | + |
| Wong 2011 | + | + | + | + | |
| Yogev 2011 | + | + | + | + | |

**Fig. 3 Risk of bias graph: review authors' judgements about each risk of bias item presented as percentages across all studies included in the meta-analyses.** Green circle with + sign, Yes, Red circle with − sign, No, Blank – not described.

such as BMI, that are easily and routinely measured in clinical practice, and thus have potential to be integrated into prediction models with the aim of achieving rapid glycaemic control. With the relatively short timeframe available to treat GDM, commencing effective therapy earlier, and thus reducing excess foetal growth, is an important target to improve outcomes. Basing treatment decisions closely on precision markers could also avoid over-medicalisation of women who are likely to achieve glucose targets with dietary counselling alone.

In our first systematic review, we identified only two studies addressing precision markers in lifestyle-based interventions for GDM, over and above the usual lifestyle intervention as standard care[19,20]. In both studies, precision markers were examined as secondary analyses of the trials and only two precision markers (communication of GWG goals according to pre-pregnancy BMI; and early GWG as a precision marker for the efficacy of technological enhancement to a behavioural intervention) were assessed; it is thus not possible to conclusively identify any precision marker in lifestyle-based interventions for GDM. This gap in the literature highlights the need for more research, as also echoed by patients and healthcare professionals participating in the 2020 James Lind Alliance (JLA) Priority Setting Partnership (PSP)[69].

Our second systematic review extends the observations of a previous systematic review reporting maternal characteristics associated with the need for insulin treatment in GDM[11]. We identified a number of additional precision markers of successful GDM treatment with lifestyle measures alone, without need for additional pharmacological therapy. The same set of predictors identified women requiring additional insulin after treatment with glyburide as with metformin, despite their different mechanisms of action. However, the numbers of women included in most studies were relatively low and most studies with data in relation to need to escalation to insulin in addition to glyburide were over 10 years old[55,56,58–60]. We acknowledge that there are also differences in diagnostic criteria, clinical practices, and preferences for choice of which drug to start as first pharmacological agent in various global regions which may limit the generalisability of our findings.

Notably, many of the identified precision markers are routinely measured in clinical practice and so could be incorporated into prediction models of need for pharmacological treatment[70,71]. By identifying those who require escalation of pharmacological therapy earlier, better allocation of resources can be achieved. Additionally, some of the precision markers identified, such as BMI, are potentially modifiable. This raises the question of how women can be helped to better prepare for pregnancy[72]. Implementing interventions prior to pregnancy could help understand if these precision markers are on the causal pathway, thus providing an opportunity for prevention and improving health outcomes.

Importantly, there was a lack of data on other potential precision treatment biomarkers, with only two eligible low-quality studies reporting maternal genetic and metabolomic findings[67,68]. In the non-pregnancy literature, efficacy of dietary interventions has been reported to differ for patients with distinct metabolic profiles, for example high fasting glucose versus high fasting insulin, or insulin resistance versus low insulin secretion[73–75]. More recent evidence from appropriately designed, prospective dietary intervention studies has confirmed that dietary interventions tailored towards specific metabolic profiles have more beneficial effects than interventions not specifically designed towards a patient's metabolic profile[76–79]. Ongoing studies such as the Westlake Precision Birth Cohort (WeBirth) in China (NCT04060056) and the USA Hoosier Moms Cohort (NCT03696368) are collecting additional biomarkers which will enhance knowledge in this field. However, implementing such measures in clinical practice, if they prove informative, could be complex and expensive and thus not suitable for use in all global contexts.

Our study has several limitations: Our reviews primarily relied on secondary analyses from observational studies that were not specifically designed to address the question of precision medicine in GDM treatment and were not powered for many of the comparisons made. Prior to introduction in clinical practice, any

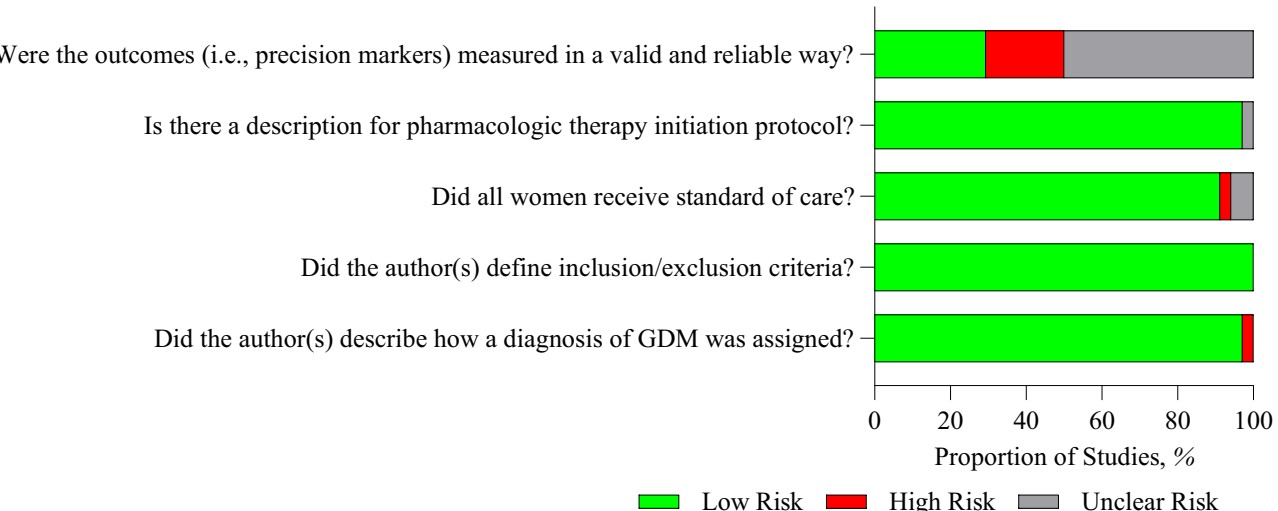

**Fig. 4 Risk of bias summary: review authors' judgements about each risk of bias item for each study included in the meta-analyses.** Green – low risk of bias, Grey – unclear risk of bias, Red – high risk of bias.

**Table 1 Lifestyle adequate to achieve target glucose levels vs need for escalation to pharmacological agent(s) to achieve glucose targets.**

| Precision Marker | Studies | Participants | Statistical Method | Effect Estimate (95%CI) | GRADE |
|---|---|---|---|---|---|
| Age (years) | 20 | 14620 | Mean difference (95%CI) | −0.98 [−1.23, −0.73] | ⊕⊕◯◯ |
| Nulliparity | 8 | 6969 | Odds Ratio (95%CI) | 1.53 [1.23, 1.89] | ⊕⊕◯◯ |
| Body mass index kg/m$^2$ | 16 | 11313 | Mean difference (95%CI) | −1.83 [−2.32, −1.35] | ⊕⊕◯◯ |
| Previous history of GDM | 13 | 9885 | Odds Ratio (95%CI) | 0.46 [0.37, 0.57] | ⊕⊕◯◯ |
| Haemoglobin A1C (%) | 8 | 4825 | Mean difference (95%CI) | −0.21 [−0.27, −0.14] | ⊕⊕◯◯ |
| Fasting glucose (mg/dl) | 13 | 8663 | Mean difference (95%CI) | −6.26 [−8.44, −4.08] | ⊕⊕◯◯ |
| 1-h glucose(mg/dl) | 10 | 6579 | Mean difference (95%CI) | −15.33 [−20.81, −9.85] | ⊕◯◯◯ |
| 2-h glucose(mg/dl) | 12 | 8255 | Mean difference (95%CI) | −9.06 [−13.55, −4.56] | ⊕◯◯◯ |
| 3-h glucose(mg/dl) | 3 | 2126 | Mean difference (95%CI) | −8.56 [−12.58, −4.54] | ⊕◯◯◯ |
| Family history of diabetes | 13 | 9256 | Odds Ratio (95%CI) | 0.66 [0.59, 0.75] | ⊕⊕◯◯ |
| Gestational age at GDM diagnosis (weeks) | 9 | 5882 | Mean difference (95%CI) | 3.06 [2.33, 3.79] | ⊕⊕◯◯ |
| Smoking history | 5 | 3488 | Odds Ratio (95%CI) | 0.80 [0.52, 1.23] | ⊕⊕◯◯ |
| Previous history of macrosomia | 7 | 5595 | Odds Ratio (95%CI) | 0.63 [0.42, 0.94] | ⊕⊕◯◯ |

Very low ⊕◯◯◯.
Low ⊕⊕◯◯.

**Table 2 Oral pharmacological agent adequate to achieve target glucose levels vs need for escalation to insulin to achieve glucose targets.**

| Precision Marker | Studies | Participants | Statistical method | Effect Estimate (95%CI) | GRADE |
|---|---|---|---|---|---|
| Age (years) | 11 | 1473 | Mean difference (95%CI) | −1.04 [−2.10, 0.03] | ⊕⊕◯◯ |
| Nulliparity | 8 | 1215 | Odds Ratio (95%CI) | 1.55 [1.17, 2.04] | ⊕⊕◯◯ |
| Body mass index (kg/m$^2$) | 10 | 1692 | Mean difference (95%CI) | −1.21 [−2.21, −0.21] | ⊕⊕◯◯ |
| Previous history of GDM | 8 | 1412 | Odds Ratio (95%CI) | 0.43 [0.30, 0.63] | ⊕⊕◯◯ |
| Haemoglobin A1C (%) | 6 | 1152 | Mean difference (95%CI) | −0.21 [−0.29, −0.13] | ⊕⊕◯◯ |
| Fasting glucose (mg/dl) | 12 | 1836 | Mean difference (95%CI) | −8.02 [−11.87, −4.16] | ⊕◯◯◯ |
| 1-h glucose (mg/dl) | 8 | 1177 | Mean difference (95%CI) | −10.64 [−18.25, −3.02] | ⊕◯◯◯ |
| 2-h glucose (mg/dl) | 10 | 1378 | Mean difference (95%CI) | −7.31 [−11.38, −3.25] | ⊕◯◯◯ |
| 3-h glucose (mg/dl) | 6 | 679 | Mean difference (95%CI) | 0.00 [−11.79, 11.79] | ⊕◯◯◯ |
| Family history of diabetes | 6 | 1040 | Odds Ratio (95%CI) | 0.79 [0.50, 1.25] | ⊕⊕◯◯ |
| Gestational age at GDM diagnosis (weeks) | 11 | 1473 | Mean difference (95%CI) | 2.64 [1.42, 3.86] | ⊕⊕◯◯ |
| Gestation at oral pharmacological agent initiation (weeks) | 7 | 967 | Mean difference (95%CI) | 3.79 [2.08, 5.51] | ⊕⊕◯◯ |

Very low ⊕◯◯◯.
Low ⊕⊕◯◯.

marker would have to be rigorously and prospectively tested with respect to sensitivity and specificity to predict treatment needs. The majority of data were extracted from clinical records leading to a lack of detail, such as the precise timing of BMI measurements, and limited information about whether BMI was self-reported or clinician measured. There was marked variation in approaches to GDM screening methods, choice of glucose challenge test and diagnostic thresholds as well as heterogeneity in glucose targets or criteria met to warrant escalation in treatment. Whilst we included studies from a range of geographical settings, the majority of studies were from high income settings, and therefore our findings may not be applicable to low- and middle-income countries. Pregnancy outcomes of precision medicine strategies for GDM also remain unknown, underscoring the need for tailored interventions that account for patient perspective and diverse patient populations.

Despite these limitations, our study has several strengths. We used robust methods to identify a broad range of precision markers, many of which are routinely measured and can be easily translated into prediction models. We excluded studies where the choice of drug was decided by the clinician based on participant characteristics to avoid bias. Our study also highlights the need for further research in this area, particularly in exploring whether there are more sensitive markers that could be identified through omics approaches.

In conclusion, our findings suggest that precision medicine for GDM treatment holds promise as a tool to stream-line individuals towards the most effective and potentially cost-effective care. Whether this will impact on short-term pregnancy outcomes and longer term health outcomes for both mother and baby is not known. More research is urgently needed to identify precision lifestyle interventions and to explore whether more sensitive markers could be identified. Prospective studies, appropriately powered and designed to allow assessment of discriminative abilities (sensitivity, specificity), and (external) validation studies are urgently needed to understand the utility and generalisability of our findings to under-represented populations. This is an area of active research with findings from ongoing studies (NCT04187521, NCT03029702, NCT05932251) eagerly awaited. Consideration of how identified markers can be implemented feasibly and cost effectively in clinical practice is also required. Such efforts will be critical for realising the full potential of precision medicine and empowering patients and their health care providers to optimise short and long-term health outcomes for both mother and child.

## Data availability
The included studies are detailed in Supplementary Data 1 and 2. The data underlying Tables 1 and 2 are in Supplementary Figs. 1–13 and 14–25, respectively. Additional information is available via contact with the corresponding author.

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

## Acknowledgements

The ADA/EASD Precision Diabetes Medicine Initiative, within which this work was conducted, has received the following support: The Covidence license was funded by Lund University (Sweden) for which technical support was provided by Maria Björklund and Krister Aronsson (Faculty of Medicine Library, Lund University, Sweden). Administrative support was provided by Lund University (Malmö, Sweden), University

of Chicago (IL, USA), and the American Diabetes Association (Washington D.C., USA). The Novo Nordisk Foundation (Hellerup, Denmark) provided grant support for in-person writing group meetings (PI: L Phillipson, University of Chicago, IL). J.M.M. acknowledges the support of the Henry Friesen Professorship in Endocrinology, University of Manitoba, Canada. N.-M.M. and R.M.R. acknowledge the support of the British Heart Foundation (RE/18/5/34216). S.E.O. is supported by the Medical Research Council (MC_UU_00014/4) and British Heart Foundation (RG/17/12/33167).

## Author contributions

All authors J.L.B., V.G., N.-M.M., J.M., J.M.Y., C.E.A., S.E.O. and R.M.R. contributed to the design of the research questions, study selection, extraction of data, data analyses, quality assessment and data interpretation. The ADA/EASD PMDI consortium members provided feedback on methodology and reporting guidelines. RMR wrote the first draft of the manuscript. All authors edited the manuscript and all approved the final version.

## Competing interests

The authors declare no competing interests.

## Additional information

## ADA/EASD PMDI

Deirdre K. Tobias[11,12], Jordi Merino[13,14,15], Abrar Ahmad[16], Catherine Aiken[8,9], Jamie L. Benham[1,198], Dhanasekaran Bodhini[17], Amy L. Clark[18], Kevin Colclough[19], Rosa Corcoy[20,21,22], Sara J. Cromer[14,23,24], Daisy Duan[25], Jamie L. Felton[26,27,28], Ellen C. Francis[29], Pieter Gillard[30], Véronique Gingras[2,3,198], Romy Gaillard[31], Eram Haider[32], Alice Hughes[19], Jennifer M. Ikle[33,34], Laura M. Jacobsen[35], Anna R. Kahkoska[36], Jarno L. T. Kettunen[37,38,39], Raymond J. Kreienkamp[14,15,23,40], Lee-Ling Lim[41,42,43], Jonna M. E. Männistö[44,45], Robert Massey[32], Niamh-Maire Mclennan[46], Rachel G. Miller[47], Mario Luca Morieri[48,49], Jasper Most[6,198], Rochelle N. Naylor[50], Bige Ozkan[51,52], Kashyap Amratlal Patel[19], Scott J. Pilla[53,54], Katsiaryna Prystupa[55,56], Sridharan Raghavan[57,58], Mary R. Rooney[51,59], Martin Schön[55,56,60], Zhila Semnani-Azad[12], Magdalena Sevilla-Gonzalez[23,24,61], Pernille Svalastoga[62,63], Wubet Worku Takele[64], Claudia Ha-ting Tam[43,65,66], Anne Cathrine B. Thuesen[13], Mustafa Tosur[67,68,69], Amelia S. Wallace[51,59], Caroline C. Wang[59], Jessie J. Wong[70], Jennifer M. Yamamoto[7,198], Katherine Young[19], Chloé Amouyal[71,72], Mette K. Andersen[13], Maxine P. Bonham[73], Mingling Chen[74], Feifei Cheng[75], Tinashe Chikowore[24,76,77,78], Sian C. Chivers[79], Christoffer Clemmensen[13], Dana Dabelea[80], Adem Y. Dawed[32], Aaron J. Deutsch[15,23,24], Laura T. Dickens[81], Linda A. DiMeglio[26,27,28,82], Monika Dudenhöffer-Pfeifer[16], Carmella Evans-Molina[26,27,28,83], María Mercè Fernández-Balsells[84,85], Hugo Fitipaldi[16], Stephanie L. Fitzpatrick[86], Stephen E. Gitelman[87], Mark O. Goodarzi[88,89], Jessica A. Grieger[90,91], Marta Guasch-Ferré[12,92], Nahal Habibi[90,91], Torben Hansen[13], Chuiguo Huang[43,65], Arianna Harris-Kawano[26,27,28], Heba M. Ismail[26,27,28], Benjamin Hoag[93,94], Randi K. Johnson[95,96], Angus G. Jones[19,97], Robert W. Koivula[98], Aaron Leong[14,24,99], Gloria K. W. Leung[73], Ingrid M. Libman[100], Kai Liu[90], S. Alice Long[101], William L. Lowe Jr.[102], Robert W. Morton[103,104,105], Ayesha A. Motala[106], Suna Onengut-Gumuscu[107], James S. Pankow[108], Maleesa Pathirana[90,91], Sofia Pazmino[109], Dianna Perez[26,27,28], John R. Petrie[110], Camille E. Powe[14,23,24,111], Alejandra Quinteros[90], Rashmi Jain[112,113], Debashree Ray[59,114], Mathias Ried-Larsen[115,116], Zeb Saeed[117], Vanessa Santhakumar[11], Sarah Kanbour[53,118], Sudipa Sarkar[53], Gabriela S. F. Monaco[26,27,28], Denise M. Scholtens[119], Elizabeth Selvin[51,59], Wayne Huey-Herng Sheu[120,121,122], Cate Speake[123], Maggie A. Stanislawski[95], Nele Steenackers[109], Andrea K. Steck[124], Norbert Stefan[56,125,126], Julie Støy[127], Rachael Taylor[128], Sok Cin Tye[129,130],

Gebresilasea Gendisha Ukke[64], Marzhan Urazbayeva[68,131], Bart Van der Schueren[109,132], Camille Vatier[133,134], John M. Wentworth[135,136,137], Wesley Hannah[138,139], Sara L. White[79,140], Gechang Yu[43,65], Yingchai Zhang[43,65], Shao J. Zhou[91,141], Jacques Beltrand[142,143], Michel Polak[142,143], Ingvild Aukrust[62,144], Elisa de Franco[19], Sarah E. Flanagan[19], Kristin A. Maloney[145], Andrew McGovern[19], Janne Molnes[62,144], Mariam Nakabuye[13], Pål Rasmus Njølstad[62,63], Hugo Pomares-Millan[16,146], Michele Provenzano[147], Cécile Saint-Martin[148], Cuilin Zhang[149,150], Yeyi Zhu[151,152], Sungyoung Auh[153], Russell de Souza[104,154], Andrea J. Fawcett[155,156], Chandra Gruber[157], Eskedar Getie Mekonnen[158,159], Emily Mixter[160], Diana Sherifali[104,161], Robert H. Eckel[162], John J. Nolan[163,164], Louis H. Philipson[160], Rebecca J. Brown[153], Liana K. Billings[165,166], Kristen Boyle[80], Tina Costacou[47], John M. Dennis[19], Jose C. Florez[14,15,23,24], Anna L. Gloyn[33,34,167], Maria F. Gomez[16,168], Peter A. Gottlieb[124], Siri Atma W. Greeley[169], Kurt Griffin[113,170], Andrew T. Hattersley[19,97], Irl B. Hirsch[171], Marie-France Hivert[14,172,173], Korey K. Hood[70], Jami L. Josefson[155], Soo Heon Kwak[174], Lori M. Laffel[175], Siew S. Lim[64], Ruth J. F. Loos[13,176], Ronald C. W. Ma[43,65,66], Chantal Mathieu[30], Nestoras Mathioudakis[53], James B. Meigs[24,99,177], Shivani Misra[178,179], Viswanathan Mohan[180], Rinki Murphy[181,182,183], Richard Oram[19,97], Katharine R. Owen[98,184], Susan E. Ozanne[185], Ewan R. Pearson[32], Wei Perng[80], Toni I. Pollin[145,186], Rodica Pop-Busui[187], Richard E. Pratley[188], Leanne M. Redman[189], Maria J. Redondo[67,68], Rebecca M. Reynolds[46], Robert K. Semple[46,190], Jennifer L. Sherr[191], Emily K. Sims[26,27,28], Arianne Sweeting[192,193], Tiinamaija Tuomi[37,136,39], Miriam S. Udler[14,15,23,24], Kimberly K. Vesco[194], Tina Vilsbøll[195,196], Robert Wagner[55,56,197], Stephen S. Rich[107] & Paul W. Franks[12,16,98,105]

[11]Division of Preventative Medicine, Department of Medicine, Brigham and Women's Hospital and Harvard Medical School, Boston, MA, USA. [12]Department of Nutrition, Harvard T.H. Chan School of Public Health, Boston, MA, USA. [13]Novo Nordisk Foundation Center for Basic Metabolic Research, Faculty of Health and Medical Sciences, University of Copenhagen, Copenhagen, Denmark. [14]Diabetes Unit, Endocrine Division, Massachusetts General Hospital, Boston, MA, USA. [15]Center for Genomic Medicine, Massachusetts General Hospital, Boston, MA, USA. [16]Department of Clinical Sciences, Lund University Diabetes Centre, Lund University, Malmö, Sweden. [17]Department of Molecular Genetics, Madras Diabetes Research Foundation, Chennai, India. [18]Division of Pediatric Endocrinology, Department of Pediatrics, Saint Louis University School of Medicine, SSM Health Cardinal Glennon Children's Hospital, St. Louis, MO, USA. [19]Department of Clinical and Biomedical Sciences, University of Exeter Medical School, ExeterDevonUK. [20]CIBER-BBN, ISCIII, Madrid, Spain. [21]Institut d'Investigació Biomèdica Sant Pau (IIB SANT PAU), Barcelona, Spain. [22]Departament de Medicina, Universitat Autònoma de Barcelona, Bellaterra, Spain. [23]Program in Metabolism and Medical & Population Genetics, Broad Institute, Cambridge, MA, USA. [24]Department of Medicine, Harvard Medical School, Boston, MA, USA. [25]Division of Endocrinology, Diabetes and Metabolism, Johns Hopkins University School of Medicine, Baltimore, MD, USA. [26]Department of Pediatrics, Indiana University School of Medicine, Indianapolis, IN, USA. [27]Herman B Wells Center for Pediatric Research, Indiana University School of Medicine, Indianapolis, IN, USA. [28]Center for Diabetes and Metabolic Diseases, Indiana University School of Medicine, Indianapolis, IN, USA. [29]Department of Biostatistics and Epidemiology, Rutgers School of Public Health, Piscataway, NJ, USA. [30]University Hospital Leuven, Leuven, Belgium. [31]Department of Pediatrics, Erasmus Medical Center, Rotterdam, The Netherlands. [32]Division of Population Health & Genomics, School of Medicine, University of Dundee, Dundee, UK. [33]Department of Pediatrics, Stanford School of Medicine, Stanford University, Stanford, CA, USA. [34]Stanford Diabetes Research Center, Stanford School of Medicine, Stanford University, Stanford, CA, USA. [35]University of Florida, Gainesville, FL, USA. [36]Department of Nutrition, University of North Carolina at Chapel Hill, Chapel Hill, NC, USA. [37]Helsinki University Hospital, Abdominal Centre/Endocrinology, Helsinki, Finland. [38]Folkhalsan Research Center, Helsinki, Finland. [39]Institute for Molecular Medicine Finland FIMM, University of Helsinki, Helsinki, Finland. [40]Department of Pediatrics, Division of Endocrinology, Boston Children's Hospital, Boston, MA, USA. [41]Department of Medicine, Faculty of Medicine, University of Malaya, Kuala Lumpur, Malaysia. [42]Asia Diabetes Foundation, Hong Kong SAR, China. [43]Department of Medicine & Therapeutics, Chinese University of Hong Kong, Hong Kong SAR, China. [44]Department of Pediatrics and Clinical Genetics, Kuopio University Hospital, Kuopio, Finland. [45]Department of Medicine, University of Eastern Finland, Kuopio, Finland. [46]Centre for Cardiovascular Science, Queen's Medical Research Institute, University of Edinburgh, Edinburgh, UK. [47]Department of Epidemiology, University of Pittsburgh, Pittsburgh, PA, USA. [48]Metabolic Disease Unit, University Hospital of Padova, Padova, Italy. [49]Department of Medicine, University of Padova, Padova, Italy. [50]Department of Pediatrics and Medicine, University of Chicago, Chicago, IL, USA. [51]Welch Center for Prevention, Epidemiology, and Clinical Research, Johns Hopkins Bloomberg School of Public Health, Baltimore, MD, USA. [52]Ciccarone Center for the Prevention of Cardiovascular Disease, Johns Hopkins School of Medicine, Baltimore, MD, USA. [53]Department of Medicine, Johns Hopkins University, Baltimore, MD, USA. [54]Department of Health Policy and Management, Johns Hopkins University Bloomberg School of Public Health, Baltimore, MD, USA. [55]Institute for Clinical Diabetology, German Diabetes Center, Leibniz Center for Diabetes Research at Heinrich Heine University Düsseldorf, Auf'm Hennekamp 65, 40225 Düsseldorf, Germany. [56]German Center for Diabetes Research (DZD), Ingolstädter Landstraße 1, 85764 Neuherberg, Germany. [57]Section of Academic Primary Care, US Department of Veterans Affairs Eastern Colorado Health Care System, Aurora, CO, USA. [58]Department of Medicine, University of Colorado School of Medicine, Aurora, CO, USA. [59]Department of Epidemiology, Johns Hopkins Bloomberg School of Public Health, Baltimore, MD, USA. [60]Institute of Experimental Endocrinology, Biomedical Research Center, Slovak Academy of Sciences, Bratislava, Slovakia. [61]Clinical and Translational Epidemiology Unit, Massachusetts General Hospital, Boston, MA, USA. [62]Mohn Center for Diabetes Precision Medicine, Department of Clinical Science, University of Bergen, Bergen, Norway. [63]Children and Youth Clinic, Haukeland University Hospital, Bergen, Norway. [64]Eastern Health Clinical School, Monash University, Melbourne, VIC, Australia. [65]Laboratory for Molecular Epidemiology in Diabetes, Li Ka Shing Institute of Health Sciences, The Chinese University of Hong Kong, Hong Kong, China. [66]Hong Kong Institute of Diabetes and Obesity, The Chinese University of Hong Kong, Hong Kong, China. [67]Department of Pediatrics, Baylor College of Medicine, Houston, TX, USA. [68]Division of Pediatric

Diabetes and Endocrinology, Texas Children's Hospital, Houston, TX, USA. [69]Children's Nutrition Research Center, USDA/ARS, Houston, TX, USA. [70]Stanford University School of Medicine, Stanford, CA, USA. [71]Department of Diabetology, APHP, Paris, France. [72]Sorbonne Université, INSERM, NutriOmic team, Paris, France. [73]Department of Nutrition, Dietetics and Food, Monash University, Melbourne, VIC, Australia. [74]Monash Centre for Health Research and Implementation, Monash University, Clayton, VIC, Australia. [75]Health Management Center, The Second Affiliated Hospital of Chongqing Medical University, Chongqing Medical University, Chongqing, China. [76]MRC/Wits Developmental Pathways for Health Research Unit, Department of Paediatrics, Faculty of Health Sciences, University of the Witwatersrand, Johannesburg, South Africa. [77]Channing Division of Network Medicine, Brigham and Women's Hospital, Boston, MA, USA. [78]Sydney Brenner Institute for Molecular Bioscience, Faculty of Health Sciences, University of the Witwatersrand, Johannesburg, South Africa. [79]Department of Women and Children's health, King's College London, London, UK. [80]Lifecourse Epidemiology of Adiposity and Diabetes (LEAD) Center, University of Colorado Anschutz Medical Campus, Aurora, CO, USA. [81]Section of Adult and Pediatric Endocrinology, Diabetes and Metabolism, Kovler Diabetes Center, University of Chicago, Chicago, USA. [82]Department of Pediatrics, Riley Hospital for Children, Indiana University School of Medicine, Indianapolis, IN, USA. [83]Richard L. Roudebush VAMC, Indianapolis, IN, USA. [84]Biomedical Research Institute Girona, IdIBGi, Girona, Spain. [85]Diabetes, Endocrinology and Nutrition Unit, Girona, University Hospital Dr Josep Trueta, Girona, Spain. [86]Institute of Health System Science, Feinstein Institutes for Medical Research, Northwell Health, Manhasset, NY, USA. [87]Department of Pediatrics, Diabetes Center, University of California at San Francisco, San Francisco, CA, USA. [88]Division of Endocrinology, Diabetes and Metabolism, Cedars-Sinai Medical Center, Los Angeles, CA, USA. [89]Department of Medicine, Cedars-Sinai Medical Center, Los Angeles, CA, USA. [90]Adelaide Medical School, Faculty of Health and Medical Sciences, The University of Adelaide, Adelaide, SA, Australia. [91]Robinson Research Institute, The University of Adelaide, Adelaide, SA, Australia. [92]Department of Public Health and Novo Nordisk Foundation Center for Basic Metabolic Research, Faculty of Health and Medical Sciences, University of Copenhagen, 1014 Copenhagen, Denmark. [93]Division of Endocrinology and Diabetes, Department of Pediatrics, Sanford Children's Hospital, Sioux Falls, SD, USA. [94]University of South Dakota School of Medicine, E Clark St, Vermillion, SD, USA. [95]Department of Biomedical Informatics, University of Colorado Anschutz Medical Campus, Aurora, CO, USA. [96]Department of Epidemiology, Colorado School of Public Health, Aurora, CO, USA. [97]Royal Devon University Healthcare NHS Foundation Trust, Exeter, UK. [98]Oxford Centre for Diabetes, Endocrinology and Metabolism, University of Oxford, Oxford, UK. [99]Division of General Internal Medicine, Massachusetts General Hospital, Boston, MA, USA. [100]UPMC Children's Hospital of Pittsburgh, Pittsburgh, PA, USA. [101]Center for Translational Immunology, Benaroya Research Institute, Seattle, WA, USA. [102]Department of Medicine, Northwestern University Feinberg School of Medicine, Chicago, IL, USA. [103]Department of Pathology & Molecular Medicine, McMaster University, Hamilton, ON, Canada. [104]Population Health Research Institute, Hamilton, ON, Canada. [105]Department of Translational Medicine, Medical Science, Novo Nordisk Foundation, Tuborg Havnevej 19, 2900 Hellerup, Denmark. [106]Department of Diabetes and Endocrinology, Nelson R Mandela School of Medicine, University of KwaZulu-Natal, Durban, South Africa. [107]Center for Public Health Genomics, Department of Public Health Sciences, University of Virginia, Charlottesville, VA, USA. [108]Division of Epidemiology and Community Health, School of Public Health, University of Minnesota, Minneapolis, MN, USA. [109]Department of Chronic Diseases and Metabolism, Clinical and Experimental Endocrinology, KU Leuven, Leuven, Belgium. [110]School of Health and Wellbeing, College of Medical, Veterinary and Life Sciences, University of Glasgow, Glasgow, UK. [111]Department of Obstetrics, Gynecology, and Reproductive Biology, Massachusetts General Hospital and Harvard Medical School, Boston, MA, USA. [112]Sanford Children's Specialty Clinic, Sioux Falls, SD, USA. [113]Department of Pediatrics, Sanford School of Medicine, University of South Dakota, Sioux Falls, SD, USA. [114]Department of Biostatistics, Johns Hopkins Bloomberg School of Public Health, Baltimore, MD, USA. [115]Centre for Physical Activity Research, Rigshospitalet, Copenhagen, Denmark. [116]Institute for Sports and Clinical Biomechanics, University of Southern Denmark, Odense, Denmark. [117]Department of Medicine, Division of Endocrinology, Diabetes and Metabolism, Indiana University School of Medicine, Indianapolis, IN, USA. [118]AMAN Hospital, Doha, Qatar. [119]Department of Preventive Medicine, Division of Biostatistics, Northwestern University Feinberg School of Medicine, Chicago, IL, USA. [120]Institute of Molecular and Genomic Medicine, National Health Research Institutes, Taipei City, Taiwan, ROC. [121]Division of Endocrinology and Metabolism, Taichung Veterans General Hospital, Taichung, Taiwan, ROC. [122]Division of Endocrinology and Metabolism, Taipei Veterans General Hospital, Taipei, Taiwan, ROC. [123]Center for Interventional Immunology, Benaroya Research Institute, Seattle, WA, USA. [124]Barbara Davis Center for Diabetes, University of Colorado Anschutz Medical Campus, Aurora, CO, USA. [125]University Hospital of Tübingen, Tübingen, Germany. [126]Institute of Diabetes Research and Metabolic Diseases (IDM), Helmholtz Center Munich, Neuherberg, Germany. [127]Steno Diabetes Center Aarhus, Aarhus University Hospital, Aarhus, Denmark. [128]University of Newcastle, Newcastle upon Tyne, UK. [129]Section on Genetics and Epidemiology, Joslin Diabetes Center, Harvard Medical School, Boston, MA, USA. [130]Department of Clinical Pharmacy and Pharmacology, University Medical Center Groningen, Groningen, The Netherlands. [131]Gastroenterology, Baylor College of Medicine, Houston, TX, USA. [132]Department of Endocrinology, University Hospitals Leuven, Leuven, Belgium. [133]Sorbonne University, Inserm U938, Saint-Antoine Research Centre, Institute of Cardiometabolism and Nutrition, Paris 75012, France. [134]Department of Endocrinology, Diabetology and Reproductive Endocrinology, Assistance Publique-Hôpitaux de Paris, Saint-Antoine University Hospital, National Reference Center for Rare Diseases of Insulin Secretion and Insulin Sensitivity (PRISIS), Paris, France. [135]Royal Melbourne Hospital Department of Diabetes and Endocrinology, Parkville, VIC, Australia. [136]Walter and Eliza Hall Institute, Parkville, VIC, Australia. [137]University of Melbourne Department of Medicine, Parkville, VIC, Australia. [138]Deakin University, Melbourne, VIC, Australia. [139]Department of Epidemiology, Madras Diabetes Research Foundation, Chennai, India. [140]Department of Diabetes and Endocrinology, Guy's and St Thomas' Hospitals NHS Foundation Trust, London, UK. [141]School of Agriculture, Food and Wine, University of Adelaide, Adelaide, SA, Australia. [142]Institut Cochin, Inserm U 10116 Paris, France. [143]Pediatric endocrinology and diabetes, Hopital Necker Enfants Malades, APHP Centre, université de Paris, Paris, France. [144]Department of Medical Genetics, Haukeland University Hospital, Bergen, Norway. [145]Department of Medicine, University of Maryland School of Medicine, Baltimore, MD, USA. [146]Department of Epidemiology, Geisel School of Medicine at Dartmouth, Hanover, NH, USA. [147]Nephrology, Dialysis and Renal Transplant Unit, IRCCS—Azienda Ospedaliero-Universitaria di Bologna, Alma Mater Studiorum University of Bologna, Bologna, Italy. [148]Department of Medical Genetics, AP-HP Pitié-Salpêtrière Hospital, Sorbonne University, Paris, France. [149]Global Center for Asian Women's Health, Yong Loo Lin School of Medicine, National University of Singapore, Singapore, Singapore. [150]Department of Obstetrics and Gynecology, Yong Loo Lin School of Medicine, National University of Singapore, Singapore, Singapore. [151]Kaiser Permanente Northern California Division of Research, Oakland, CA, USA. [152]Department of Epidemiology and Biostatistics, University of California San Francisco, San Francisco, CA, USA. [153]National Institute of Diabetes and Digestive and Kidney Diseases, National Institutes of Health, Bethesda, MD, USA. [154]Department of Health Research Methods, Evidence, and Impact, Faculty of Health Sciences, McMaster University, Hamilton, ON, Canada. [155]Ann & Robert H. Lurie Children's Hospital of Chicago, Department of Pediatrics, Northwestern University Feinberg School of Medicine, Chicago, IL, USA. [156]Department of Clinical and Organizational Development, Chicago, IL, USA. [157]American Diabetes Association, Arlington, VA, USA. [158]College of Medicine and Health Sciences, University of Gondar, Gondar, Ethiopia. [159]Global Health Institute, Faculty of Medicine and Health Sciences, University of Antwerp, 2160 Antwerp, Belgium. [160]Department of Medicine and Kovler Diabetes Center, University of Chicago, Chicago, IL, USA. [161]School of Nursing, Faculty of Health Sciences, McMaster University, Hamilton, ON, Canada. [162]Division of Endocrinology, Metabolism, Diabetes, University of Colorado, Boulder, CO, USA. [163]Department of Clinical Medicine, School of Medicine, Trinity College Dublin, Dublin, Ireland, UK.

[164]Department of Endocrinology, Wexford General Hospital, Wexford, Ireland, UK. [165]Division of Endocrinology, NorthShore University HealthSystem, Skokie, IL, USA. [166]Department of Medicine, Prtizker School of Medicine, University of Chicago, Chicago, IL, USA. [167]Department of Genetics, Stanford School of Medicine, Stanford University, Stanford, CA, USA. [168]Faculty of Health, Aarhus University, Aarhus, Denmark. [169]Department of Pediatrics and Medicine and Kovler Diabetes Center, University of Chicago, Chicago, USA. [170]Sanford Research, Sioux Falls, SD, USA. [171]University of Washington School of Medicine, Seattle, WA, USA. [172]Department of Population Medicine, Harvard Medical School, Harvard Pilgrim Health Care Institute, Boston, MA, USA. [173]Department of Medicine, Universite de Sherbrooke, Sherbrooke, QC, Canada. [174]Department of Internal Medicine, Seoul National University College of Medicine, Seoul National University Hospital, Seoul, Republic of Korea. [175]Joslin Diabetes Center, Harvard Medical School, Boston, MA, USA. [176]Charles Bronfman Institute for Personalized Medicine, Icahn School of Medicine at Mount Sinai, New York, NY, USA. [177]Broad Institute, Cambridge, MA, USA. [178]Division of Metabolism, Digestion and Reproduction, Imperial College London, London, UK. [179]Department of Diabetes & Endocrinology, Imperial College Healthcare NHS Trust, London, UK. [180]Department of Diabetology, Madras Diabetes Research Foundation & Dr. Mohan's Diabetes Specialities Centre, Chennai, India. [181]Department of Medicine, Faculty of Medicine and Health Sciences, University of Auckland, Auckland, New Zealand. [182]Auckland Diabetes Centre, Te Whatu Ora Health New Zealand, Auckland, New Zealand. [183]Medical Bariatric Service, Te Whatu Ora Counties, Health New Zealand, Auckland, New Zealand. [184]Oxford NIHR Biomedical Research Centre, University of Oxford, Oxford, UK. [185]University of Cambridge, Metabolic Research Laboratories and MRC Metabolic Diseases Unit, Wellcome-MRC Institute of Metabolic Science, Cambridge, UK. [186]Department of Epidemiology & Public Health, University of Maryland School of Medicine, Baltimore, MD, USA. [187]Department of Internal Medicine, Division of Metabolism, Endocrinology and Diabetes, University of Michigan, Ann Arbor, MI, USA. [188]AdventHealth Translational Research Institute, Orlando, FL, USA. [189]Pennington Biomedical Research Center, Baton Rouge, LA, USA. [190]MRC Human Genetics Unit, Institute of Genetics and Cancer, University of Edinburgh, Edinburgh, UK. [191]Yale School of Medicine, New Haven, CT, USA. [192]Faculty of Medicine and Health, University of Sydney, Sydney, NSW, Australia. [193]Department of Endocrinology, Royal Prince Alfred Hospital, Sydney, NSW, Australia. [194]Kaiser Permanente Northwest, Kaiser Permanente Center for Health Research, Portland, OR, USA. [195]Clinial Research, Steno Diabetes Center Copenhagen, Herlev, Denmark. [196]Department of Clinical Medicine, Faculty of Health and Medical Sciences, University of Copenhagen, Copenhagen, Denmark. [197]Department of Endocrinology and Diabetology, University Hospital Düsseldorf, Heinrich Heine University Düsseldorf, Moorenstr. 5, 40225 Düsseldorf, Germany.

