## [Peer Review File · Communications Medicine]

Reviewers' comments:

Reviewer #1 (Remarks to the Author):

This works summaries findings from two systematic reviews with a view to determining whether a precision-based medicine approach could be adopted for gestational diabetes. The theory behind this is an important one and while no major conclusions can be drawn for this work in relation to some markers, it highlights the gaps in the literature together with the avenues that could be explored in determining effective precision based approaches. The methodology adopted for this is robust and clearly

Major Comments

1. The inclusion of the two studies examining different precision approaches to behavioural interventions appears at odds with the main purpose of the commentary as defined by the authors by nature of neither of the studies examining whether the approaches facilitated achievement of glucose targets (knowledge gap identified page 3, summary page 6). The scope of the aims should be expanded if these trials are to be included. A comment on the outcomes reflecting degree of glycaemic control would also be useful e.g. macrosomia/ neonatal hypoglycaemia. While a recent study has suggested that women whose pregnancies are complicated by GDM and who have appropriate GWG have a reduced risk of LGA neonates (Mustaniemi et al 2021), the evidence base relating to levels of appropriate GWG particularly in the context of GDM are limited and the limitations of targeting GWG should be explored in the discussion.

Minor Comments

1. While many important limitations are covered, the authors do not include factors such as heterogeneity in glucose targets or criteria met to warrant escalation in treatment. A comment on the potential contribution of ethnicity should also be made.

Reviewer #2 (Remarks to the Author):

Review: Precision Gestational Diabetes Treatment: systematic review and meta-analyses, Reynolds et al. for Communications Medicine.

Thank you for asking me to review this systematic review and meta-analysis which I very much enjoyed reading. In summary the authors have systematically reviewed the literature for gestational diabetes precision markers that might predict effective lifestyle and pharmacological therapy as part of a wider Precision Medicine in Diabetes Initiative (PMDI). As such, the findings and perhaps more importantly the paucity of evidence identified will act as an important springboard for the gestational diabetes community.

The statistical approach appears robust and reproducible, it is clear that undertaking this SR and MA was a mammoth task and I commend the authors.

I have several minor comments and observations:

Introduction:

1. In the penultimate paragraph of the introduction, I wondered whether 'receiving oral agents' should actually say 'receiving pharmacological agents'? Or perhaps it is in reference to escalating beyond oral therapy? Either way, this sentence would benefit from clarification.
2. At the beginning of the last sentence of the footnote, should 'consequent' say 'consistent'?

Results:

3. 'Precision markers of need for escalation of pharmacological interventions. etc' reads clunkily here and elsewhere as the title of the second review – can this be improved? Maybe just 'precision markers for escalation of...?'
4. I was a little confused by the 4th paragraph starting 'There were 34 studies...' and how this related to the previous and subsequent paragraphs. I think it might be helpful if for example, you added, 'Of these, there were 34 studies...' to clarify that you are still talking about the second review. Or perhaps these 3 subsequent paragraphs all talking about the 2nd review might be coalesced into 1 paragraph?
5. Can you confirm where Huhtala et al. exactly fits in this review? It does seem to fulfill eligibility based on the published PROSPERO (factors that predict responses to pharmacological interventions) but doesn't seem to quite fit in the second described review 'precision markers of need for escalation of pharmacological interventions' as unless I am mistaken, I can see no data on those that needed to escalate in this secondary analysis?
(Should it be the former, might this similar paper have a place in the review? Metabolic phenotyping by treatment modality in obese women with gestational diabetes suggests diverse pathophysiology: An exploratory study: <https://doi.org/10.1371/journal.pone.0230658>)
6. Figure 2.1: there is a typo in the first line of this Figure 'how a diagnose of GDM...'
7. Synthesis of results: end of first paragraph – 'Neither study...' Do you mean that neither study was designed specifically to examine whether a precision approach worked? As it is written I read it as not examining it at all, yet the info has been identified within the study.
8. Precision markers of need for escalation section: Can you rephrase the first sentence of the second paragraph? 'Table 2 and Supplementary Figures 1.1 – 1.3...' as currently it is a little difficult to understand.
9. Paragraph 3: Is n=1669 correct? The meta analysis of fasting glucose which included all 12 of the studies has an n of 1836 (Table 3) – if I have interpreted correctly?
10. Is the sensitivity analysis between metformin and glyburide data shown somewhere in the manuscript? If so, please reference, if not please note that the data is not shown.
11. Supplementary Table 1 – the refs in the lefthand column are repeated for 30, 37 and 39

Discussion:

12. I am aware of a couple of ongoing studies of precision treatment that you might wish to reference:

HINT-GDM – Heterogeneity informed Nutrition Therapy for GDM:

(<https://clinicaltrials.gov/ct2/show/NCT04187521>)

MATCH-GDM – Metabolic analysis for treatment choice in GDM

(<https://clinicaltrials.gov/ct2/show/NCT03029702>)

Response to reviewers COMMSMED-23-0334

Reviewer #1:

We thank Reviewer 1 for their supportive comments and are happy to address the points raised.

Major Comments

1. The inclusion of the two studies examining different precision approaches to behavioural interventions appears at odds with the main purpose of the commentary as defined by the authors by nature of neither of the studies examining whether the approaches facilitated achievement of glucose targets (knowledge gap identified page 3, summary page 6). The scope of the aims should be expanded if these trials are to be included. A comment on the outcomes reflecting degree of glycaemic control would also be useful e.g. macrosomia/ neonatal hypoglycaemia. While a recent study has suggested that women whose pregnancies are complicated by GDM and who have appropriate GWG have a reduced risk of LGA neonates (Mustaniemi et al 2021), the evidence base relating to levels of appropriate GWG particularly in the context of GDM are limited and the limitations of targeting GWG should be explored in the discussion.

Response: Reviewer 1 was concerned about the scope of the lifestyle review. We agree with the reviewer that the stated objective of achievement of glucose targets is not answered by the inclusion of these two behavioural studies. However, in our registered protocol on PROSPERO, the goal of the lifestyle systematic review was to determine characteristics or factors that predict responses to personalized behavioural interventions. We were looking at the effect of any behavioural intervention in addition to standard of care on any maternal or neonatal outcome and were not looking specifically for achieving glucose targets. We have broadened the aims listed in the introduction to reflect the registered PROSPERO protocol, which was more inclusive than just glycaemic targets.

To address the reviewer's comment on outcomes reflecting degree of glycaemic control, we have edited the sentence in the synthesis of results: "*Neither study examined whether a precision approach to specific lifestyle interventions facilitated achievement of glucose targets during pregnancy.*" by adding "...or improved outcomes that reflect glycaemic control during pregnancy such as macrosomia, large for gestational age, or neonatal hypoglycaemia."

We note the reviewer's suggestion to discuss the limitations of targeting GWG in the discussion. The goal of our review was not to evaluate whether the maternal and neonatal outcomes reported for GDM are appropriate or not so we have not made this addition. However we are happy to take editorial guidance if they think such a discussion is appropriate.

Minor Comments

1. While many important limitations are covered, the authors do not include factors such as heterogeneity in glucose targets or criteria met to warrant escalation in treatment. A comment on the potential contribution of ethnicity should also be made.

Response: many thanks for these helpful suggestions. We have added a comment about the heterogeneity in glucose targets and criteria met to warrant escalation in treatment to the limitations. We had already stated that we were unable to draw conclusions about the potential contribution of ethnicity in the results section (page 7) so have not commented further on this.

Reviewer #2:

We thank the reviewer for their supportive comments and in particular noting “it is clear that undertaking this SR and MA was a mammoth task and I commend the authors.” Reviewer 2 had only minor comments and noted some typographical errors which we have corrected.

1. In the penultimate paragraph of the introduction, I wondered whether ‘receiving oral agents’ should actually say ‘receiving pharmacological agents’? Or perhaps it is in reference to escalating beyond oral therapy? Either way, this sentence would benefit from clarification.

Response: this sentence has been clarified.

2. At the beginning of the last sentence of the footnote, should ‘consequent’ say ‘consistent’?

Response: this has been corrected

3. ‘Precision markers of need for escalation of pharmacological interventions. etc’ reads clunkily here and elsewhere as the title of the second review – can this be improved? Maybe just ‘precision markers for escalation of...?’

Response: We thank the reviewer for this helpful suggestion - we have removed ‘of need’ to simplify and have edited this throughout the manuscript.

4. I was a little confused by the 4th paragraph starting ‘There were 34 studies...’ and how this related to the previous and subsequent paragraphs. I think it might be helpful if for example, you added, ‘Of these, there were 34 studies...’ to clarify that you are still talking about the second review. Or perhaps these 3 subsequent paragraphs all talking about the 2nd review might be coalesced into 1 paragraph?

Response: We have coalesced this section into 1 paragraph which we hope adds clarity.

5. Can you confirm where Huhtala et al. exactly fits in this review? It does seem to fulfil eligibility based on the published PROSPERO (factors that predict responses to pharmacological interventions) but doesn’t seem to quite fit in the second described review ‘precision markers of need for escalation of pharmacological interventions’ as unless I am mistaken, I can see no data on those that needed to escalate in this secondary analysis?

(Should it be the former, might this similar paper have a place in the review? Metabolic phenotyping by treatment modality in obese women with gestational diabetes suggests diverse pathophysiology: An exploratory study: <https://doi.org/10.1371/journal.pone.0230658>)

Response: We thank the reviewer for questioning this. Huhtala et al was included as in the supplementary data tables there is regression analysis data linking some lipid measurements to birth weight data in the metformin and insulin treated groups. We have therefore modified the introduction, re-stating the aims to more accurately reflect our PROSPERO registration which was very broad, and included maternal and neonatal outcomes as well as glucose targets. The other paper that the reviewer refers to was identified in our search, but was excluded on the basis that the sample size – for the analyses we were interested in – was less than 50: 71 people with GDM of which 28 had diet alone, 20 had metformin and 23 were on insulin.

6. Figure 2.1: there is a typo in the first line of this Figure ‘how a diagnose of GDM...’

Response: We apologise for this error - this has been corrected.

7. Synthesis of results: end of first paragraph – ‘Neither study...’ Do you mean that neither study was designed specifically to examine whether a precision approach worked? As it is written I read it as not examining it at all, yet the info has been identified within the study.

Response: We have edited this sentence (also to address comments from reviewer 1)

8. Precision markers of need for escalation section: Can you rephrase the first sentence of the second paragraph? ‘Table 2 and Supplementary Figures 1.1 – 1.3...’ as currently it is a little difficult to understand.

Response: We have rephrased this sentence so that hopefully it is easier to understand.

9. Paragraph 3: Is n=1669 correct? The meta analysis of fasting glucose which included all 12 of the studies has an n of 1836 (Table 3) – if I have interpreted correctly?

Response: Many thanks for spotting this error. There was a typographical error and the data in the table was correct and this has now been corrected.

10. Is the sensitivity analysis between metformin and glyburide data shown somewhere in the manuscript? If so, please reference, if not please note that the data is not shown.

Response: we have added that the data are not shown.

11. Supplementary Table 1 – the refs in the lefthand column are repeated for 30, 37 and 39

Response: We apologise for this error - we have corrected this typo

12. I am aware of a couple of ongoing studies of precision treatment that you might wish to reference:

HINT–GDM – Heterogeneity informed Nutrition Therapy for GDM:

(<https://clinicaltrials.gov/ct2/show/NCT04187521>)

MATCh-GDM – Metabolic analysis for treatment choice in GDM

(<https://clinicaltrials.gov/ct2/show/NCT03029702>)

Response: Many thanks for recommending including these trials in the discussion. We have added these to the discussion together with ToPMedDiP-Towards Precision Medicine for Diabetes in Pregnancy (<https://clinicaltrials.gov/study/NCT05932251>)

REVIEWERS' COMMENTS:

Reviewer #1 (Remarks to the Author):

Thank you for addressing the comments raised.

Reviewer #2 (Remarks to the Author):

Thank you for addressing all of my comments.

I think you have changed all occurrences in the text of 'precision markers of need for escalation...' to 'precision markers for escalation' (which reads much better) except just one in the 'Precision markers for escalation of pharmacological interventions to achieve glucose targets in GDM' section - sentence commencing 'Twelve studies...

Apart from that, from my point of view, it is good to go.